# Efficacy of Valganciclovir Treatment Depends on the Severity of Hearing Dysfunction in Symptomatic Infants with Congenital Cytomegalovirus Infection

**DOI:** 10.3390/ijms20061388

**Published:** 2019-03-19

**Authors:** Shohei Ohyama, Ichiro Morioka, Sachiyo Fukushima, Keiji Yamana, Kosuke Nishida, Sota Iwatani, Kazumichi Fujioka, Hisayuki Matsumoto, Takamitsu Imanishi, Yuji Nakamachi, Masashi Deguchi, Kenji Tanimura, Kazumoto Iijima, Hideto Yamada

**Affiliations:** 1Departments of Pediatrics, Kobe University Graduate School of Medicine, Kobe 650-0017, Japan; ohyama@med.kobe-u.ac.jp (S.O.); sachi4@med.kobe-u.ac.jp (S.F.); k-yamana@kjf.biglobe.ne.jp (K.Y.); xitian1410@yahoo.co.jp (K.N.); siwatani2011@gmail.com (S.I.); fujiokak@med.kobe-u.ac.jp (K.F.); iijima@med.kobe-u.ac.jp (K.I.); 2Department of Pediatrics and Child Health, Nihon University School of Medicine, Tokyo 173-8610, Japan; 3Departments of Clinical Laboratory, Kobe University Hospital, Kobe 650-0017, Japan; hisa-mt@live.jp (H.M.); imanisi@med.kobe-u.ac.jp (T.I.); nakamati@med.kobe-u.ac.jp (Y.N.); 4Departments of Obstetrics and Gynecology, Kobe University Graduate School of Medicine, Kobe 650-0017, Japan; deguchi@med.kobe-u.ac.jp (M.D.); tanimurakenji@gmail.com (K.T.); yhideto@med.kobe-u.ac.jp (H.Y.)

**Keywords:** auditory brainstem response, congenital cytomegalovirus infection, hearing dysfunction, severity, valganciclovir

## Abstract

Although earlier studies have shown that antiviral treatment regimens using valganciclovir (VGCV) improved hearing function in some infants with congenital cytomegalovirus (CMV) infection; its efficacy on the severity of hearing dysfunction is unclear. We conducted a prospective study among 26 infants with congenital CMV infections from 2009 to 2018. Oral VGCV (32 mg/kg/day) was administered for 6 weeks (November 2009 to June 2015; *n* = 20) or 6 months (July 2015 to March 2018, *n* = 6). Hearing function was evaluated by measuring the auditory brainstem response before VGCV treatment and at 6 months. Hearing dysfunction, defined as a V-wave threshold >40 dB, was categorized into: most severe, ≥91 dB; severe, 61–90 dB; and moderate, 41–60 dB. Hearing improvement was defined as a decrease of ≥20 dB from the pretreatment V-wave threshold. Of 52 ears in 26 infants with congenital CMV infection, 29 (56%) had hearing dysfunction, and of 29 ears, 16 (55%) improved after VGCV treatment. Although, 16 (84%) of 19 ears with moderate or severe hearing dysfunction improved after treatment (*p* < 0.001), 10 ears with the most severe form did not. In conclusion, VGCV treatment is effective in improving moderate and severe hearing dysfunction in infants with congenital CMV infection.

## 1. Introduction

Congenital sensorineural hearing loss can be caused by hereditary or non-hereditary factors. Although the former is caused by gene mutations such as *SLC26A4* and *GJB2*, the latter most frequently originates from congenital cytomegalovirus (CMV) infection. A study in the United States (US) reported that congenital CMV infection accounted for 21% of congenital hearing loss [1]. The use of antiviral therapies such as intravenous (IV) ganciclovir (GCV) or oral valganciclovir (VGCV) for congenital CMV infection, has garnered clinical interest in recent years because various reports have shown the effects of these therapies on the reversal of hearing loss or prevention of its deterioration [2,3,4,5]. However, the potential associations between the baseline of hearing dysfunction and the efficacy of antiviral therapy have not been adequately investigated. To our knowledge, the only study on this topic was conducted by Bilavsky et al. in 2016 [5], who retrospectively compared the hearing status of infants with congenital CMV infection involving central nervous abnormalities at baseline to their hearing status at ≥1 year of age. The study followed one of two regimens of antiviral therapy involving either: 1) 6 weeks of IV GCV (10 mg/kg/day), which was followed by oral VGCV at the doses of 34 mg/kg/day for 6 weeks and then 17 mg/kg/day until the age of 1. Or 2) Oral VGCV (34 mg/kg/day) for 12 weeks, and subsequent doses of 17 mg/kg/day until the age of 1. They observed that the likelihood of improvement of hearing loss post-treatment was inversely associated with its severity at baseline; with improvement observed in 78%, 68%, and 41% of ears with mild, moderate, and severe hearing dysfunction, respectively [5]. However, their work had several limitations, including its retrospective nature, and the fact that there was only a small proportion of infants with hearing impairment at baseline (36%). Furthermore, the improvement or deterioration of hearing function was defined as a ≥10 dB decrease or increase in the threshold of the auditory brainstem response (ABR); this result indicates that some obtained data may have inter-rater errors. Regimen-related issues included the mixture of GCV and VGCV in a treatment group, and the choice of a long, one-year course, which has limited supporting evidence. Finally, whilst baseline hearing was consistently assessed at the same time across subjects, the exact timing of post-treatment assessments (at age ≥1 year) varied from child to child.

To address these limitations, we decided to determine the therapeutic efficacy of oral VGCV monotherapy (32 mg/kg/day) by using a prospective design, comparing a 6-week regimen with a 6-month regimen, with a uniform timing of 6 months after birth for the post-treatment hearing assessment. To minimize confounders due to inter-rater error and methodology, we defined hearing improvement or deterioration more strictly as a respective decrease or increase of ≥20 dB in the wave V threshold of the ABR. Our objective was to conduct the first prospective study to determine if the efficacy of VGCV is dependent on the severity of hearing dysfunction in symptomatic infants with congenital CMV infection.

## 2. Results

### 2.1. Patient Characteristics

From 2009 to 2018, 26 symptomatic infants with congenital CMV infection received VGCV therapy at Kobe University Hospital (6-week group, *n* = 20; 6-month group, *n* = 6). Table 1 shows their clinical characteristics. The median gestational age and birth weight were 37 weeks and 2268 g, respectively. All the subjects were symptomatic, and the most frequent signs were brain image abnormalities (*n* = 23, 88%) and abnormal ABR (*n* = 21, 81%). Adverse events were observed in 11 children (42%); with neutropenia as the most common, affecting 10 (38%) of the subjects. Adverse events occurred in 10/20 (50%) and 1/6 (17%) of infants in the 6-week and 6-month group, respectively (*p* = 0.33). Neutropenia, which led to a temporary discontinuation of treatment was observed in 8 children (31%); 5 (19%) required granulocyte colony-stimulating factor (G-CSF), 3 (12%) continued at half-dosage (16 mg/kg/day), and 1 (4%) required the drug change to foscarnet. No infant with neutropenia developed any additional severe bacterial infection. Additionally, the blood and urine CMV loads were reduced from baseline by VGCV therapy in 25/26 infants (96%). At the end of treatment, residual CMV DNA was detected in the blood and urine of 3 (12%) and 6 (23%) infants, respectively. Both cases belonged to the 6-week group; neither viremia nor viruria was detected in the 6-month group (blood: *p* = 0.56, urine: *p* = 0.28).

### 2.2. Baseline Results of ABRs and Classification Based on the Hearing Severity

Based on the baseline ABR (*n* = 26 patients, 52 ears), 29 ears (56%) were classified as abnormal (most severe: 10, severe: 6, moderate: 13), and 23 (44%) was considered non-abnormal (44%; mild: 8, normal: 15). 

### 2.3. Efficacy of VGCV Treatment on Hearing

Figure 1 presents the observed treatment effects for abnormal and non-abnormal ears. Hearing function improved in 16 (55%) and was maintained in 11 (38%) of 29 abnormal ears; however, deterioration was observed in 2 (7%). In comparison, the hearing function was maintained in 20 (87%) but worsened in 3 (13%) of 23 non-abnormal ears. 

### 2.4. Differences in Therapeutic Efficacy Due to VGCV Treatment Duration

We compared the differences in therapeutic efficacy observed between the two VGCV regimens (6-week vs. 6-month), and found no statistically significant differences in efficacy as follows (*p* = 1.00); the hearing function was improved or maintained with the same degree in 22 (92%) of 24 abnormal ears in the 6-week regimen and 5 (100%) of 5 abnormal ears in the 6-month regimen. Furthermore, the hearing function was maintained in 14 (88%) of 16 non-abnormal ears in the 6-week regimen and 6 (86%) of 7 non-abnormal ears in the 6-month regimen. Hearing deterioration was observed in 4 children (5 ears: 2 abnormal ears in the 6-week regimen, and 2 and 1 non-abnormal ears in the 6-week and the 6-month regimen, respectively) across both groups; residual CMV was detected in the urine of 1/4, and the blood of 0/4.

### 2.5. Number of Improved Ears after VGCV Treatment Based on Hearing Severity in Abnormal Ears

Out of 29 abnormal ears, 10 ears with the “most severe” form of hearing dysfunction at baseline did not show significant improvement (0%); however, hearing improved in 5 (83%) of 6 ears with “severe” hearing dysfunction (*p* < 0.01 vs. most severe) and in 11 (85%) of 13 ears with baseline “moderate” hearing dysfunction (*p* < 0.001 vs. most severe). 

## 3. Discussion

In this prospective study, hearing dysfunction due to congenital CMV infection improved after VGCV treatment in 55% of abnormal ears. Although the “most severe” form of hearing dysfunction was unlikely to respond to the drug, it can potentially improve hearing function in ears with severe and moderate impairments. Conversely, some ears with normal hearing function at baseline showed signs of deterioration after VGCV therapy.

In a systematic review of hearing dysfunction associated with congenital CMV infection, Goderis et al. found that 32.8% of cases with infections exhibited hearing impairment at birth [6]. Similarly, Bilavsky et al. reported the incidences of congenital hearing impairment in 36.2% of the cohort of infants with symptomatic congenital CMV infection in a single-center study [5]. In our cohort, hearing impairment including mild forms of impairment was noted in one or both ears of 21/26 children (81%). Our study seems to have enrolled more infants with congenital CMV infection with hearing dysfunction than that documented in previous reports [5,6]. This can be explained by the following factor. Kobe University Hospital manages severe fetuses with CMV infection, who already exhibited some symptoms in utero, such as abdominal effusion or brain image abnormalities on ultrasound.

Oral VGCV therapy was associated with adverse events in 11 children (42%), and neutropenia was observed in 10 (38%). Kimberlin et al. observed neutropenia in 29/46 (63%) of their cohort treated with IV GCV [2]. Whitley et al. also reported a similarly high incidence in infants during IV GCV therapy (29/47, 62%) [7]. However, oral VGCV regimens may be less likely to trigger neutropenia than IV GCV according to a more recent study by Kimberlin et al. which recorded neutropenia in only 19–21% of infants given oral VGCV at a dosage of 32 mg/kg/day [4]. At the end of treatment, we detected residual CMV DNA in the blood and urine of 3 (12%) and 6 (23%) infants, respectively; while Lombardi et al. detected residual CMV DNA in the serum and urine of 4/12 (33%) and 6/12 (50%) of subjects, respectively, at the end of a six-week VGCV regimen. However, most infants with viremia exhibited viral load reductions compared with baseline levels [8]. This seems to corroborate our observation of reduced CMV load in 96% of our cohort (25/26). The sole case where CMV load increased following treatment was the same infant who was suspected of a GCV-resistant strain and was treated with foscarnet. One of the adverse events unrelated to myelosuppression (besides neutropenia and thrombocytopenia) was hypocalcemia; however, the evidence for a causal relationship with VGCV is unclear.

The study by Bilavsky et al. [5] showed observed the reversal of hearing in 50/77 ears (64.9%). Conversely, we observed an improvement in 55% of our samples of “abnormal” ears, indicating a lower efficacy in comparison with Bilavsky’s study. Potential reasons for this may include our stricter definition of improvement as wave-V threshold decrease of >20 dB, and the higher incidence of the more severe form of hearing dysfunction at baseline in our cohort. Furthermore, excluding the 10 ears with the “most severe” form, hearing dysfunction was worsened in 5/42 ears (12%) in our cohort; this incidence is somewhat higher than that recorded by Bilavsky et al., who observed a deterioration of hearing function in only 6/276 ears (2.2%) [5]. Again, this was likely because our cohort exclusively included severe cases. The different treatment duration might be affected. In 2003, Kimberlin et al. reported that following a 6-week GCV regimen, 5/24 of infants (21%) included in their cohort had worsened hearing 6 months after birth than at baseline [2]. Although they did not compare subgroups based on the initial severity, their findings seemed to corroborate our findings. After VGCV therapy was completed, we detected residual viruria in only one of 4 infants whose hearing function had deteriorated. Lombardi et al. [8] also observed CMV viremia and viruria in 50% and 33% of infants, respectively after VGCV treatment; however, none had any deterioration in hearing. Purported associations of hearing dysfunction with CMV loads in the blood and urine are controversial. Lanari et al. found that neonatal blood CMV load correlated with the likelihood of developing sequelae at 12 months of age [9]. Marsico et al. reported the deterioration of hearing function to be uncommon in infants who consistently tested negative for CMV viremia from 2 weeks through 6 months after commencing treatment [10]. In contrast, Ross et al. found no correlation between blood CMV load and hearing dysfunction in a cohort of 135 infants with congenital CMV infection [11].

VGCV treatment duration was not associated with differential treatment effects in our cohort. Kimberlin et al.’s study in 2015 [4], a randomized control trial comparing a 6-week and 6-month regimen of oral VGCV, resulted in similar findings. Although hearing function at 12- and 24-months follow-up was significantly improved or maintained in the 6-month group, no such difference was evident between the groups at the 6-month time point.

In our cohort, whereas hearing function improved in 84% of ears with “severe” or “moderate” impairment at baseline, no such reversals were observed among ears rated “most severe” (0%). A lot remains uncertain about the pathogenic mechanism of congenital CMV infection-induced hearing loss [6]. However, previous studies have hypothesized on the ability of the virus to damage endolymphatic structures and the stria vascularis, causing potassium disequilibrium and, consequently, structural degeneration of the nerves [12]. It is still unclear why antiviral therapy improves hearing dysfunction in some cases; however, reducing endolymphatic CMV load may restore potassium regulation structures, which may regenerate the endolymphatic potential [13]. Our findings suggest that hearing dysfunction is somewhat reversible in cases of moderate or severe impairment, who may benefit from antiviral therapy. However, this plasticity may already be lost in the “most severe” cases, leaving little hope for improvement. These trends mirror those of Bilavsky et al. [5], who reported an inverse relationship between baseline severity of hearing dysfunction and the probability of improvement once children reach 1 year of age. Moreover, their group observed an improved hearing in 40.9% of ears with “severe,” hearing dysfunction which was defined as a wave-V threshold of ≥75 dB. Our stricter definitions of the “most severe” form of hearing dysfunction (≥90 dB) and “improvement” could frustrate simple comparisons.

Our study had several limitations:
(1)Treatment duration was modified midway through the study period in July 2015, from 6 weeks of oral VGCV (32 mg/kg/day) to 6 months. However, ethical considerations about choosing the optimal treatment option for our patients dictated this decision, based on the evidence presented by Kimberlin et al. in 2015 supporting the superiority of 6-month versus 6-week regimens of VGCV [4]. As indicated in a review article by Rawlinson et al. [14], antiviral courses longer than 6 months are not yet currently recommended, given the paucity of evidence that infants receive any additional benefit. This led us to conclude that the 6-month regimen was best supported by the current consensus in the field.(2)Our decision to measure our main outcome at 6 months after birth could only capture the relatively short-term effects. It is critical for future studies to pursue longer-term follow-up; this would enable the assessment of any deterioration or progressive deafness. However, we achieved our original goal of assessing VGCV’s effects at a consistent time point for all patients. Hearing assessments during long-term follow-up can be affected by other factors besides CMV infection, such as otitis media. Therefore, we believe our choice of 6 months after birth as the assessment time point was appropriate for focusing on VGCV’s efficacy in treating congenital CMV infection.(3)Despite the impressive length of our study, spanning around 9 years, our subgroups were relatively small after classifying them by hearing severity. We consider this final limitation as characteristic of any single-center prospective studies for this subject with congenital CMV infection.

## 4. Materials and Methods

### 4.1. Study Design and Patients

From November 2009 to March 2018, a single-center prospective study was conducted at Kobe University Hospital to investigate the efficacy of VGCV treatment on hearing dysfunction in infants with symptomatic congenital CMV infection. Because antiviral treatment is off-label for congenital CMV infection, we fully explained the risks and benefits of its use to the legal guardians of each subject, and their written consent was obtained before starting treatment. All subjects with congenital CMV infection were authorized to undergo VGCV therapy. This study was conducted in accordance with the Declaration of Helsinki, and the protocol was approved by the Ethics Committee of Kobe University Graduate School of Medicine (no. 1214). 

### 4.2. Diagnosing Congenital CMV Infection

Congenital CMV infection was definitively diagnosed based on the following methods:
(1)A positive result of a filter paper-based screening to detect CMV DNA in urine, a method developed previously by our team [15] or a presence of clinical symptoms and findings of congenital CMV infection;(2)a positive result of real-time quantitative PCR for CMV DNA in the urine as a confirmed diagnosis, as previously established [16]. Diagnoses were made within 3 weeks of birth in all cases [15].

### 4.3. Definition of Symptomatic Congenital CMV Infection

Following a confirmed diagnosis, symptoms were determined based on the findings of the physical exams, hematological testing, chest X-rays, abdominal ultrasound, fundoscopy, ABR testing, and brain imaging (ultrasound, computed tomography, and magnetic resonance imaging). Congenital CMV infection was considered “symptomatic” if one or more of the following symptoms were present: microcephaly, hepatosplenomegaly/hepatitis, thrombocytopenia, brain imaging abnormalities, eye complications such as retinal choroiditis, and abnormal ABR [16,17,18]. Microcephaly was defined as a head circumference ≥-1.5 standard deviation, compared to the average for Japanese neonates at the corresponding gestational age [19]. Hepatosplenomegaly was diagnosed based on abdominal ultrasound or X-ray images. Hepatitis was defined as serum aspartic aminotransferase ≥100 U/L, and thrombocytopenia was defined as platelet count <100,000/µL. Additionally, the imaging diagnosis of brain abnormalities, which were identified as intracranial calcification, ventricular dilatation, cortical dysplasia, or white matter injuries, was performed by a radiologist blinded to the child’s clinical course. A pediatric ophthalmologist diagnosed retinal choroiditis by using fundoscopy.

### 4.4. Measuring CMV Viral Load in Blood and Urine

Using a previously reported method [16], DNA was extracted from the peripheral blood and urine using QIAamp DNA Mini kits (Qiagen Corp., Tokyo, Japan). CMV-DNA copy number (CN) was measured using real-time quantitative PCR. CMV DNA in the blood was expressed in terms of CN per 10^6^ white blood cells, with a negative cut-off of 1 × 10^2^ CN/10^6^ white blood cells. CMV DNA in urine was expressed in terms of CN/mL, with a negative cut-off of 3 × 10^3^ CN/mL [16].

### 4.5. Definition of Hearing Dysfunction

ABR was measured using the Neuropack S1 system (Nihon Kohden Co., Tokyo, Japan). First, we tested the ability of the stimuli at amplitudes 90 dB, 60 dB, and 30 dB to evoke ABR. Hearing status was classified as “normal” if the auditory stimulus evoked a detectable wave V at 30 dB. However, if wave V was undetectable at a given loudness, the test was repeated with a stimulus 10 dB higher in amplitude. The wave V threshold was defined as the lowest amplitude to evoke the ABR. Based on the classification of the Japan Audiological Society, hearing dysfunction was defined as a V-wave threshold and categorized as: most severe, ≥91 dB; severe, 61–90 dB; moderate, 41–60 dB; 31–40 dB, mild; and normal, ≤30 dB. The abnormal ears were classified as: most severe, severe, or moderate for analysis; while the non-abnormal ears were classified as mild or normal [20].

### 4.6. VGCV Treatment Protocols

Symptomatic infants with congenital CMV infection were hospitalized for the first 6 weeks of VGCV therapy to monitor drug effects and adverse events. Depending on their conditions, the patients were released to continue treatment in an outpatient setting after 6 weeks. The subjects enrolled from November 2009 to June 2015 were administered 6 weeks of oral VGCV (32 mg/kg/d), while those enrolled from July 2015 onward were administered the same dose for 6 months [3,4,8]. The infants were regularly monitored for treatment effects and adverse events. A neonatologist examined the infants daily during the first 6 weeks of treatment, and their blood and urine CMV loads were measured once a week during the first 6 weeks of treatment and at least once monthly thereafter. Neutropenia (i.e., neutrophil count <500/mm^3^) is a common side effect of VGCV [2]; when observed, doctors temporarily discontinued treatment and waited for the neutrophil count to recover before resuming treatment. In this event, the duration of treatment (either 6 weeks or 6 months) was measured from the resumption of VGCV administration. If this took for more than one week, G-CSF was administered, or some subjects resumed treatment at a reduced dosage, based on the attending physician’s discretion. Furthermore, the medications were substituted to foscarnet if drug-resistant CMV was suspected during the treatment course.

### 4.7. Efficacy Assessment

Wave-V threshold was determined before the start of treatment (baseline) and 6 months after birth using the method mentioned above. Hearing improvement/exacerbation was defined as a ≥20 dB decrease/increase in this value from baseline to the sixth-month time point; hearing maintenance was defined when the wave-V threshold was the same, or the change was within 20 dB [20,21]. 

Our primary outcomes were the improvement rates in the abnormal ear group and each severity group of hearing dysfunction at baseline. Our secondary outcome was the rate at which hearing was maintained in the non-abnormal ear group.

### 4.8. Statistical Analysis

Data were organized in 2 × 2 contingency tables and statistically analyzed using Fisher’s exact test. *p* < 0.05 was considered significant.

## 5. Conclusions

We concluded that VGCV treatment for either 6 weeks or 6 months is effective in improving hearing dysfunction and maintaining the hearing degree in infants with congenital CMV infection, except for those with the most severe form. Subsequently, further studies using multiple center-based prospective cohorts will be needed to confirm our results.

## Figures and Tables

**Figure 1 ijms-20-01388-f001:**
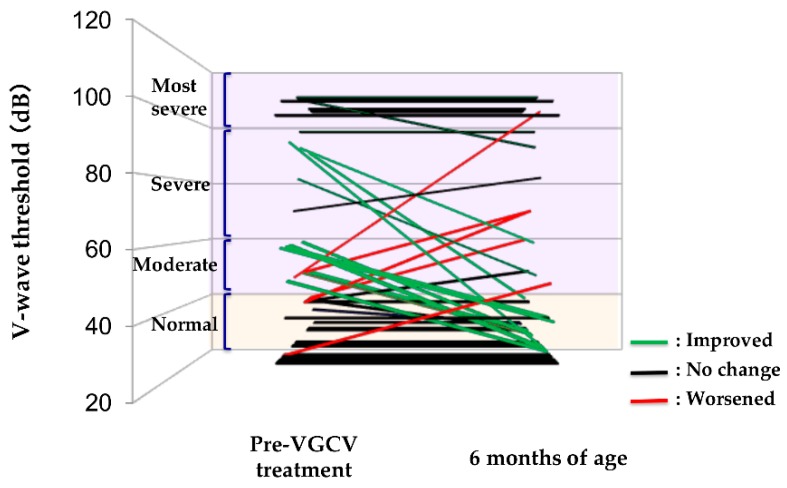
Efficacy of V-wave threshold after VGCV treatment. VGCV, valganciclovir.

**Table 1 ijms-20-01388-t001:** Clinical background.

Symptomatic Infants with Congenital CMV Infection	*n* = 26
Gestational age, weeks	37 (31–40)
Birth weight, g	2268 (940–3312)
Male	8 (31)
Any symptoms	26 (100)
Microcephaly	8 (31)
Hepatosplenomegaly/hepatitis	10 (38)
Thrombocytopenia	12 (46)
Brain image abnormality	23 (88)
Eye complications	7 (27)
Hearing dysfunction	21 (81)
Duration of VGCV treatment	
6 weeks/6 months	20 (77)/6 (23)
Age when treatment started, days after birth	12 (4–105)
Adverse event of VGCV therapy ^1^	11 (42)
Neutropenia	10 (38)
Thrombocytopenia	2 (8)
Genital bleeding	1 (4)
Impetigo	1 (4)
Hypocalcemia	1 (4)
CMV load in blood before VGCV treatment, copies/10^6^ WBC	7.15 × 10^2^ (2.2 × 10^2^–1.7 × 10^5^)
CMV load in urine before VGCV treatment, copies/mL	5.25 × 10^7^ (1.9 × 10^4^–2.4 × 10^9^)
Residual CMV in blood at the time when VGCV treatment finished	3 (12)
Residual of CMV in urine at the time when VGCV treatment finished	6 (23)

^1^ Three infants had multiple side effects. Data are shown as median (range) or number (percent). CMV, cytomegalovirus; VGCV, valganciclovir; WBC, white blood cell.

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
