# Peer review of "Efficacy of Valganciclovir Treatment Depends on the Severity of Hearing Dysfunction in Symptomatic Infants with Congenital Cytomegalovirus Infection"

_ijms, 2019, doi:10.3390/ijms20061388_

Reviewer 1 Report

This manuscript is marginal with respect to originality but does address an important issue. I would therefore recommend publication as a Brief Communication or a Letter.

1.     Table 1 is adequately presented in the text. Fig 1 and Table 2 can be deleted and replaced with better annotation to Figure 1. Table 3 can be better presented in the text….this has been done briefly so further detail can be applied easily. The same applies to Table 4.  This reduces the data to one figure and perhaps 1 Table…drawn from Tables 1 and 4.

2.     The Introduction and Discussion should be cut to remove repetition…eg: re refs 2 and 5.

Author Response

Response to Reviewer #1 comment:

This manuscript is marginal with respect to originality but does address an important issue. I would therefore recommend publication as a Brief Communication or a Letter.

Response:

We discussed with the editor and this paper will be published as a Communication if accepted. Our revised manuscript has been re-edited by Editage (www.editage.jp)regarding English language editing. We will send the editor the certificate of English language editing from Editage.

1.     Table 1 is adequately presented in the text. Fig 1 and Table 2 can be deleted and replaced with better annotation to Figure 1. Table 3 can be better presented in the text….this has been done briefly so further detail can be applied easily. The same applies to Table 4.  This reduces the data to one figure and perhaps 1 Table…drawn from Tables 1 and 4.

Response1:

We thank you for suggestions. As the reviewer suggested, Figure 1 and Tables 2 to 4 on the original manuscript were deleted, and we described the results in text in the revised manuscript. Finally, one figure and one table are remained in the revised manuscript. 

2.     The Introduction and Discussion should be cut to remove repetition…eg: re refs 2 and 5. 

Response 2:

We alsothank you for suggestions. The repeated sentences were deleted in the revised manuscript.

Reviewer 2 Report

The manuscript "Efficacy of valganciclovir treatment depends on the severity of hearing dysfunction in symptomatic infants with congenital cytomegalovirus infection" by Ohyamaet al., analyze the correlation between valganciclovir treatment and hearing dysfunction in infants with congenital HCMV infection.

Given the clinical significance of HCMV infection and the lack of a standardize treatment, these data provide an important progress for the management of congenital infected newborns.

I think that the analyses are very detailed and in my opinion the paper can be published without substantial modifications.

Author Response

Response to Reviewer #2 comment:

The manuscript "Efficacy of valganciclovir treatment depends on the severity of hearing dysfunction in symptomatic infants with congenital cytomegalovirus infection" by Ohyamaet al., analyze the correlation between valganciclovir treatment and hearing dysfunction in infants with congenital HCMV infection. Given the clinical significance of HCMV infection and the lack of a standardize treatment, these data provide an important progress for the management of congenital infected newborns. I think that the analyses are very detailed and in my opinion the paper can be published without substantial modifications.

Response:

We thank you for your warm comments to our paper. Our revised manuscript has been re-edited by Editage (www.editage.jp)regarding English language editing. We will send the editor the certificate of English language editing from Editage.

Round  2

Reviewer 1 Report

The introduction is arguably still over long but the manuscript is much improved